# Polysaccharides from *Holothuria leucospilota* Relieve Loperamide-Induced Constipation Symptoms in Mice

**DOI:** 10.3390/ijms24032553

**Published:** 2023-01-29

**Authors:** Ziqi Wang, Yali Shi, Shiyu Zeng, Yuanping Zheng, Huaijie Wang, Haihui Liao, Jie Song, Xinyue Zhang, Jun Cao, Chuan Li

**Affiliations:** 1Key Laboratory of Food Nutrition and Functional Food of Hainan Province, School of Food Science and Engineering, Hainan University, Haikou 570228, China; 2Collaborative Innovation Center of Provincial and Ministerial Co-Construction for Marine Food Deep Processing, Dalian Polytechnic University, Dalian 116034, China

**Keywords:** *Holothuria leucospilota* polysaccharides, constipation, hormone, short-chain fatty acid, gut microbiota

## Abstract

A vital bioactive component of marine resources is *Holothuria leucospilota* polysaccharides (HLP). This study examined whether HLP could regulate intestinal flora to treat loperamide-induced constipation. Constipated mice showed signs of prolonged defecation (up by 60.79 min) and a reduced number of bowel movements and pellet water content (decreased by 12.375 and 11.77%, respectively). The results showed that HLP treatment reduced these symptoms, reversed the changes in related protein expression levels in the colon, and regulated the levels of active peptides associated with the gastrointestinal tract in constipated mice, which significantly improved water-electrolyte metabolism and enhanced gastrointestinal motility. Meanwhile, it was found that intestinal barrier damage was reduced and the inflammatory response was inhibited through histopathology and immunohistochemistry. As a means to further relieve constipation symptoms, treatment with low, medium, and high HLP concentrations increased the total short-chain fatty acid (SCFA) content in the intestine of constipated mice by 62.60 μg/g, 138.91 μg/g, and 126.51 μg/g, respectively. Moreover, an analysis of the intestinal flora’s gene for 16S rRNA suggested that the intestinal microbiota was improved through HLP treatment, which is relevant to the motivation for the production of SCFAs. In summary, it was demonstrated that HLP reduced loperamide-induced constipation in mice.

## 1. Introduction

As a typical intestinal disorder, constipation is characterized by dry stools, prolonged defecation time, and difficult defecation [1]. It increases the prevalence of developing coronary heart disease and ischemic stroke, which negatively affects patients’ lives and public health resources [2,3]. The occurrence of constipation is caused by colonic movement disorders, defecation disorders, or adverse drug reactions [4]. Gut microbiota have a crucial impact on human health, which has a close relationship with reshaping the integrity of the intestinal barrier, providing nutrition and energy for the human body, regulating host immunity [5,6,7], etc. These factors make it a significant part of modern biomedical research. In recent studies, the intestinal microbiota structure of individuals with constipation differed from that of healthy individuals, revealing that an imbalance of intestinal microbiota may be one of the risk factors for constipation [8].

Currently, the main therapeutic methods for constipation are biofeedback treatment, increasing fluid and dietary fiber intake, prescription laxatives, and surgical treatment [9,10,11]. However, most clinic constipation drugs have poor therapeutic and toxic side effects [12].

*Holothuria leucospilota* is a kind of tropical sea cucumber widely distributed in shallow reef areas in the waters around Hainan, Guangdong, Guangxi, and Taiwan provinces of China and the Indo-Western Pacific Ocean. It contains anticoagulant, anticancer, anti-infective, immunostimulatory, and metabolism-regulating active ingredients [13]. *Holothuria leucospilota* contains saponins, polysaccharides, peptides, and other bioactive components [14]. Especially as a vital bioactive component of marine resources, *Holothuria leucospilota* polysaccharides (HLP) have the functions of reducing blood lipids, promoting fermentation, alleviating the symptoms of type 2 diabetes and hyperlipidemia, and improving the gut microbiome and immunity [14,15,16,17,18,19]. Over the past few years, scholars from various countries have extracted many bioactive polysaccharides from various organisms, which are ideal base materials for drugs and healthy food. Meanwhile, researchers also found that many polysaccharides had positive effects on treating constipation. For example, polysaccharides extracted from *Durio zibethinus Murr rind* relieve constipation and maintain intestinal flora balance [20]. *Chrysanthemum morifolium* polysaccharides extracted from *Chrysanthemum morifolium Ramat* treat constipation by improving the gut bacterial ratio between beneficial and harmful bacteria [21]. However, the mechanism by which HLP alleviates constipation by regulating intestinal flora is still not completely clear.

Based on our findings in this study, we hypothesized that HLP regulated the intestinal flora to alleviate loperamide-induced constipation. The mice constipation index, intestinal metabolite content, intestinal hormone levels, intestinal pathology sections, and 16S rRNA sequencing were studied using molecular biology approaches to verify the critical role of intestinal microbiota in the HLP treatment of constipation.

## 2. Results and Discussion

### 2.1. Effects of HLP on the Parameters Related to Defecation in Constipated Mice

Measurements of changes in constipation-related parameters were used to evaluate the effects of HLP on constipation in mice. In comparison with the NC group, the constipation mice showed obvious symptoms of weight loss, a reduced number of feces, and prolonged defecation time (Figure 1A–C, *p* < 0.05), which were the same symptoms as constipated mice in previous studies [22]. However, a significant increase of 1.95 g in body weight was observed in the HLP-L group, and the fecal weight in the HLP-M group also increased by 97.16% of the dry weight and 36.39% of the wet weight (Figure 1D,E, *p* < 0.05). At the same time, the constipation mice treated with HLP had a significant increase in pellet water content (Figure 1F, *p* < 0.05) (HLP-L, HLP-M, and HLP-H groups were 49.51 ± 2.07%, 50.42 ± 3.60%, and 54.28 ± 5.47%, respectively). Among them, the HLP-M and HLP-H groups even outperformed the NC group (50.35 ± 2.40%). In this experiment, HLP treatment significantly reduced the time to defecation and increased the number, weight, and water content of stools in mice. Previous studies have shown *Chrysanthemum morifolium* polysaccharides to have a relieving effect on constipation [21]. We hypothesized that HLP could promote intestinal peristalsis and water metabolism, maintain the integrity of the colon, regulate the relevant physiological indicators, and optimize the composition of intestinal flora, thus achieving a therapeutic effect on constipation. The next experiments were conducted to test these speculations.

### 2.2. Effects of HLP on the Morphology of Colon Tissue in Constipated Mice

By observing the H&E-stained colonic tissue sections (Figure 2), the cellular arrangement of colonic tissue in constipated mice (MC group) was disrupted, and the muscle layer was thinned. In previous studies, the colonic villi of constipated mice were also found to be loosely arranged or even broken, and there was a decrease in mucosal and muscular layer thickness [23,24]. In contrast, HLP treatment improved abnormal morphology in the colon and restored the integrity of the ileal epithelium in constipated mice to varying degrees. The function of HLP in repairing intestinal epithelial integrity in loperamide-induced constipation mice was demonstrated in colonic histomorphology. The intestinal epithelium is a selective physical barrier essential to intestinal homeostasis. Its disruption can lead to imbalances in the intestinal microbiota and disruption of the metabolism, which can cause inflammation and intestinal disease [25]. In this experiment, HLP maintained the homeostasis of the intestinal environment by protecting the intestinal epithelium from rupture and thickening the muscle layer of the colonic wall, achieving the purposes of maintaining intestinal microbial balance and water and electrolyte metabolic balance. Therefore, the occurrence of intestinal diseases could be prevented, while constipation was also relieved.

### 2.3. Effects of HLP on Regulatory Peptides Related to the Gastrointestinal Tract in Constipation Mice

As peptides related to gastrointestinal regulation in the intestinal tract, hormones adjust the motility, secretion, and absorption of the digestive system through multiple pathways. In comparison to the NC group, constipated mice had significantly lower levels of gastrin (GAS) and substance P (SP) hormones in their guts (Figure 3A,B, *p* < 0.05), while there was a profound increase in somatostatin (SS) and vasoactive intestinal peptide (VIP) levels (Figure 3C,D, *p* < 0.05). HLP treatment significantly reversed these changes (*p* < 0.05). In comparison with the MC group, the treatment effects were most obvious in the HLP-M group, where the levels of SP and GAS increased by 14.60 ng/L and 7.36 ng/L, respectively. Additionally, the levels of SS and VIP decreased by 3.33 ng/L and 36.16 ng/L, respectively.

SS, SP, GAS, and VIP are all essential regulators of gastrointestinal motility. The neurotransmitter SS has inhibitory properties that suppress hormones in secretory cells. VIP relaxes smooth muscle and reduces gastrointestinal motility [21,26]. Their content is directly proportional to the severity of constipation symptoms. An endogenous peptide called SP plays a role in immune regulation and attenuating intestinal injury [27], and GAS stimulates gastric acid secretion [28]. The elevated levels of these two hormones contribute to the suppression of the inflammatory response, maintaining intestinal barrier integrity, and promoting the production of fatty acids. In the gut of constipated mice, changes in all four hormones inhibited intestinal motility and slowed gastrointestinal metabolism, which correlated with prolonged defecation time and reduced constipation weight. The results confirmed that HLP treatment could regulate the secretion of gastrointestinal hormones to alleviate signs of constipation. In the previous study, constipation could be relieved by *Durio zibethinus rind* polysaccharides by increasing GAS and SP contents while decreasing SS contents [20], which was similar to the HLP in this experiment.

### 2.4. Effects of HLP on the Intestinal Metabolites of Constipated Mice

Short-chain fatty acids (SCFAs) are the main intestinal microbial metabolites produced by the glycolysis of colonic flora. This study examined the effects of HLP on intestinal microbial metabolites by measuring the content of acetic acid, propionic acid, butyric acid, valeric acid, isobutyric acid, isovaleric acid, and total acid in each group of mice. In comparison with the MC group, constipated mice treated with HLP had significantly higher SCFA levels (Figure 4, *p* < 0.05), with the most pronounced effect in the HLP-H group, which significantly elevated the content of almost all acids (propionic acid, butyric acid, isobutyric acid, valeric acid, isovaleric acid, and total acid in the HLP-H group were 76.21 ± 3.12 μg/g, 139.90 ± 14.76 μg/g, 66.25 ± 2.30 μg/g, 76.85 ± 0.92 μg/g, 103.55 ± 10.64 μg/g, and 579.18 ± 15.22 μg/g, respectively, *p* < 0.05).

SCFAs inhibited intestinal inflammation [29], which was clearly corroborated by the colon tissue sections in this experiment. They played a significant role in relieving constipation by enhancing electrolyte and water metabolism [30], which increased fecal water content in the constipation mice. In addition, the peptides associated with gastrointestinal regulation were promoted by SCFAs [24]. All these accelerated intestinal motilities in constipated mice, thus increasing the amount of stool and reducing the time to pass the first black stool, which was consistent with the results of this experiment. In this study, adding HLP provided the raw material for glycolysis by the intestinal flora, increasing the level of SCFAs. Thus, it was demonstrated that HLP could indirectly alleviate constipation symptoms by regulating SCFA content.

### 2.5. Effects of HLP on the Expression of Water Metabolism and Motility of Intestinal Tract-Related Genes and Proteins in Constipated Mice

The degrees of expression of AQP3, AQP4, AQP9, SCF, and c-Kit were all significantly decreased in constipated mice while increasing markedly after HLP treatment (Table 1, *p* < 0.05). Meanwhile, the changes were more obvious in the HLP-L and HLP-M groups.

Based on immunohistochemistry data, the expression levels of AQP9 and c-Kit, which were consistent with the above statistics, were assessed at the protein level. The brown color was a positive expression in the images, and the brown area increased significantly after HLP treatment. It was found that there was a significant reduction in the mean optical density (IOD) values of the MC group (10.88 for AQP9 and 5.06 for c-Kit) compared with that of the NC group (23.92 for AQP9 and 12.89 for c-Kit) (Figure 5A,B, *p* < 0.05). These values significantly rebounded after HLP treatment (*p* < 0.05). AQP9 levels were even higher than those in the NC group after HLP treatments (HLP-L, HLP-M, and HLP-H groups were 25.35 ± 2.34, 24.09 ± 4.92, and 25.04 ± 2.15, respectively).

In the intestinal tract, AQPs play a vital role in the reabsorption and secretion of water and electrolytes [31]. The lack of AQP expression in the intestine in this experiment limits water transport across cellular pathways, leading to the reduction of fecal water content and the development of inflammation in constipated mice, which may also be a significant cause of constipation in mice. HLP promoted the balance of water and electrolyte transport and inhibited the inflammatory response in the intestine by upregulating the expression level of AQPs, thus playing an essential role in alleviating constipation.

In the gastrointestinal tract, Interstitial Cells of Cajal (ICCs) expressing c-Kit receptor tyrosine kinase serve as pacemaker cells and mediators of neuromuscular transmission [32,33]. Reduced or absent ICC has been found in the colon of constipated patients [34]. Stem cell factor (SCF) binds to c-Kit. The SCF/c-Kit system regulated intestinal motility, inflammation, and nerve growth [35]. The decrease in the expression of SCF and c-Kit in constipated mice weakened intestinal motility and slowed intestinal peristalsis, resulting in longer defecation times and reduced fecal number and fecal weight, which is consistent with the symptoms exhibited in constipation mice in this study. In contrast, HLP increased intestinal motility and improved loperamide-induced constipation by increasing SCF and c-Kit expression levels. Therefore, HLP relieved constipation in mice by regulating the expression of water metabolism and motility of intestinal tract-related genes and proteins, an effect that was also demonstrated with beverages rich in fructooligosaccharide [12].

### 2.6. Effects of HLP on the Diversity and Community Structure of Gut Microbiota in Constipated Mice

As constipation may be caused by an imbalance of gut microbiota, we analyzed the 16S rRNA gene sequences of gut microbiota in six experimental groups to determine their composition and diversity. After filtering out ineligible sequences from 42 stool samples, 1,664,142 valid sequences were aggregated using the RDP classifier Bayesian algorithm for OTU representative sequences with a 97% similarity level. Different taxonomic levels were applied to quantify the diversity and composition of the gastrointestinal microbial community.

In this study, we used the alpha diversity of intestinal flora to analyze species richness and equitability. The Chao index could be used to estimate the richness of the flora in the sample, and the Shannon index reflected the species diversity within the sample. The Chao index decreased in constipated mice (MC group) and rebounded after HLP treatment (Figure 6B). In contrast, the Shannon index did not change significantly (Figure 6A), indicating that the intestinal flora diversity of mice was only moderately affected by loperamide-induced constipation. The results suggested that mice with constipation benefited from HLP treatment by having an abundance of intestinal flora. It regulated the intestinal microbial ecological balance and alleviated the symptoms of constipation.

At the phylum level (Figure 6C), *Firmicutes* and *Bacteroidetes* were the two dominant taxa, accounting for more than 90% of all bacteria. In the MC group’s intestines, there was an increase in *Firmicutes* and *Campilobacterota* but a decrease in *Bacteroidetes*. *Alginate* oligosaccharides maintained the mucosal barrier by increasing the abundance of *Firmicutes* in the mice’s intestines [36]. A stable internal environment is associated with the *Firmicutes* to *Bacteroidetes* ratio [37]. In comparison to the NC group (1.98), the changes in the ratio in the MC group (2.23) may lead to various diseases. HLP treatment improved this situation, which regulated the ratio of *Firmicutes* to *Bacteroidetes* (HLP-L and HLP-H groups were 1.79 and 1.84, respectively) to maintain the flora balance in the intestine.

A higher abundance of intestinal flora was found in *norank_f_Muribaculaceae* and *lactobacillus* at the genus level (Figure 6D), which was also positively correlated with fatty acid content [38,39]. Meanwhile, HLP treatment enriched the abundance of *norank_f_Muribaculaceae*, consistent with the effects of other polysaccharides on intestinal flora [40].

At the family level (Figure 6E), the modulatory effects of HLP on the intestinal bacterial structure were explored by heat map analysis, which showed 30 significantly different families in abundance in the six experimental groups. *Lachnospiraceae*, *Lactobacillaceae*, and *Muribaculaceae* were the dominant families in the intestinal flora of mice. The abundance of *Muribaculaceae* and *Ruminococcaceae* decreased in the intestine of mice in the MC group, while the abundance of *Helicobacteraceae* and *Lactobacillaceae* increased. *Ruminococcaceae* and *Muribaculaceae* promoted the production of short-chain fatty acid salts [41], which was consistent with the changes in the content of SCFAs measured in this experiment. Moreover, in humans and animals, *Helicobacteraceae* was found to be associated with diseases, such as inflammatory bowel disease [42], which was consistent with the phenomena observed in histopathological sections of the colon of constipated mice. In addition, the close species composition of the HLP-H and PC groups to that of the NC group in the heat map suggested that both high-dose HLP and drug treatment reversed the imbalance of intestinal flora in constipated mice.

Therefore, HLP treatment modulated the composition of the intestinal microbiota to promote SCFA production, thus stimulating the level of positive hormone, decreasing negative hormone, and upregulating the expression of water channel proteins, SCF, and c-Kit to improve constipation (Figure 7).

## 3. Materials and Methods

### 3.1. Material and Reagents

Dried *Holothuria leucospilotas* used in the experiment were locally sourced (Haikou, Hainan, China). Professor Yongqin Feng from Hainan University identified the species. Loperamide was purchased from Janssen Pharmaceutical Company (Xi’an, Shaanxi, China). Kits for enzyme-linked immunosorbent assays (Elisa) were purchased from Xinyu Biotechnology Company (Shanghai, China). Analytical purity was achieved for all other chemicals.

### 3.2. Extraction and Purification of HLP

HLP was purified according to the previous study with minor modifications [43]. The dried *Holothuria leucospilota* were soaked in distilled water for 24 h at 4 °C, the viscera and coelomic membrane were removed, and the mixture was crushed. Subsequently, sea cucumbers were hydrolyzed with papain, and crude polysaccharides were precipitated with cetylpyridine chloride and ethanol. Then, the product was decolorized with activated carbon, deproteinized with trichloroacetic acid, and steamed until the solution was thick. After three days of dialysis with distilled water, the product was freeze-dried to obtain HLP with high purity. The whole process took 210 h with an extraction rate of about 4%.

### 3.3. Animals and Experimental Design

60 female KM mice (6 weeks old) without specific pathogens, weighing 18–22 g, were purchased from Hunan Slac Jingda Laboratory Animal Co., Ltd. (Changsha, Hunan, China), license key SCXK (Xiang) 2019-0004. After adaptive feeding for one week at 22 ± 1 °C, mice were randomly sorted into a total of six groups based on their body weight, with 10 mice in each group: the low-dosage HLP group (HLP-L group, loperamide with 50 mg/kg of HLP), the middle-dosage HLP group (HLP-M group, loperamide with 100 mg/kg of HLP), the high-dosage HLP group (HLP-H group, loperamide with 200 mg/kg of HLP), the normal control group (NC group, sterile normal saline solution), the model control group (MC group, loperamide), and the positive control group (PC group, loperamide with polyethylene glycol 4000 powder). Mice were kept in different cages according to groups.

The mouse constipation model was established by slightly modifying the method of the previous study [44]. A sterile normal saline solution was gavaged to mice in the NC group; the other five groups received injections with 10 mg/kg BW loperamide (dissolved in sterile saline) once a day for one week. The constipation model was successfully established when reduced fecal water content and particle number were observed in other mice except for the NC group. Then, 50 mg/g, 100 mg/g, and 200 mg/g of BW HLP solution were gavaged to the HLP-L group, HLP-M group, and HLP-H group, respectively, and 3 g/kg of BW polyethylene glycol 4000 powder was gavaged to the PC group. The four groups were treated for a total of seven days. The NC and MC groups were treated with the same volume of sterile saline daily. During the experiment, all mice were given free water and food.

After collecting the mouse feces, about 0.2 mL of blood was collected using the orbital venous plexus method. The blood samples were centrifuged at 4000× *g* for 15 min to obtain serum after 1 h of resting. Using 10% chloral hydrate, mice were anesthetized, dislocated, and executed. The serum, gastrointestinal tissues, and feces were stored at −80 °C for further study.

### 3.4. Analyses of Constipation-Related Parameters

During the experimental period, the mice’s bodies were weighed daily to observe whether there were significant changes. After the seventh day of drug treatment, all mice were deprived of food for 12 h and then gavaged with 2 mL of ink. After that, time to pass the first black pellet and the amount of black stool pellets excreted within 3 h of each mouse were counted. The stools at this time were weighed as the wet weight of feces, M1 (g). Then, the stools were dried in an electric oven at 105 °C until their weight remained constant. Weight at this time was defined as the dry weight of feces, M2 (g). The ratio of the changes in dry weight to the wet weight of feces was used to calculate the fecal water content:Fecal water content (%) = [M1 − M2]/M1 × 100

### 3.5. Histopathology and Immunohistochemistry

Using a microtome (Leica, Shanghai, China), colon tissues were cut into 4-mm-thick sections after being fixed in 4% paraformaldehyde solution and embedded in paraffin. The sections were dewaxed to water, stained by hematoxylin and eosin (H&E), then dehydrated and sealed in the neutral gel [21]. The histopathological morphology of colon sections of mice in six groups was observed by light microscopy (Nikon, Tokyo, Japan) and photographed.

The paraffin-embedded colon sections were dewaxed, rehydrated, antigen repaired, and incubated overnight at 4 °C by adding stem cell factor (c-Kit) antibodies (Rabbit, GB11073-2, Servicebio, Wuhan, China) and aquaporin-9 (AQP9) antibodies (Rabbit, A8540, ABclonal, Wuhan, China). After that, the sections were washed with PBS, adding an HRP-conjugated goat anti-rabbit IgG. Subsequently, the sections were colored using the DAB kit, re-stained with hematoxylin, and finally dehydrated and sealed. The above reagents (in addition to the AQP9 antibodies) were purchased from Servicebio Technology Company (Wuhan, China). Representative photomicrographs were acquired by light microscopy with a Nikon DS-U3 imaging system. The mean IOD of colonic tissues was counted by Image-Pro Plus 6.0 software to assess the AQP9 and c-Kit expression levels.

### 3.6. Enzyme-Linked Immunosorbent Assay (ELISA)

By adding antibodies, antigens, and enzyme-labeled antibodies to the microtiter plate, ELISA kits use the double antibody sandwich method to form a triple complex. After thorough washing, TMB was added to the plate for color development. Then a microplate reader (Kaiao, Beijing, China) was used to measure the absorbance (OD value) of the complexes at 450 nm. To analyze the changes of gastrointestinal hormones in mouse serum, the absorbance (OD value) was substituted into the standard curve to calculate the values of vasoactive intestinal peptide (VIP, XY9M1447), substance P (SP, XY9M1589), gastrin (GAS, XYM9444051), and growth inhibitor (SS, XYM904931).

### 3.7. RNA Extraction and Real-Time qPCR (RT-qPCR)

The Eastep^®^ Super Total RNA Extraction Kit (LS1040, Promega, Beijing, China) was used to extract total RNA from liquid nitrogen-ground small intestinal tissues. The purity of RNA was measured by a microspectrophotometer (Colibri, Titertek Berthold, Germany), with an admissive OD260/OD280 value ranging from 1.8 to 2.0. Then the First Strand cDNA Synthesis Kit (KR118, Tiangen, Beijing, China) was used to synthesize cDNA by reverse transcription. For RT-qPCR, the extension of cDNA was completed with the SuperReal Preix Plusreagent Kit (SYBR Green) (FP201, Tiangen, Beijing, China) on the CFX Connect Real-time PCR system (BioRad, Hercules, CA, USA). The primer sequences used in this experiment to assess the levels of the related genes are shown in Table 2, covering c-Kit, its ligand stem cell factor (SCF), and water channel proteins (AQP3, AQP4, and AQP9). The relative differences in gene expression levels in the intestine of each group of mice were counted by the 2^−ΔΔCt^ (Livak) method, using the measured glycerol triphosphate dehydrogenase (GADPH) as the reference gene.

### 3.8. Quantitative Assay of Short-Chain Fatty Acids (SCFAs)

The SCFA content in fecal samples was calculated on the basis of preceding studies with minor modifications [15]. After freeze-drying mouse feces, a 50 mg fecal sample was taken from each group. Then 500 µL of a saturated sodium chloride solution was added to the sample and fully homogenized. The samples were acidified by sulfuric acid, and SCFAs were extracted by n-hexane. Shaking and centrifuging were used to collect the supernatant (4000× *g*, 4 °C, 15 min). As intestinal metabolites, the content of SCFAs in mice was measured by the GC system (7890A, Agilent Technologies, Santa Clara, CA, USA).

### 3.9. Analysis of Microbial Diversity

After extracting bacterial genomic DNA from each group of collected mouse fecal samples, the V3-V4 variable region of 16S rRNA was amplified using primers 338F (5′-ACTCCTACGGGAGGCAGCAG-3′) and 806R (5′-GGACTACHVGGGTWTCTAAT-3′). After that, the PCR products were sequenced on the Illumina MiSeq-PE250 platform to count each sample’s colony composition and structure at the taxonomic levels of phylum, family, and genus.

### 3.10. Statistical Analysis

The statistics were presented as mean ± standard error of the mean (SEM). Each group of independent samples was compared for significant differences using IBM SPSS Statistics 22.0 (SPSS Inc., Chicago, IL, USA), and a *p* value < 0.05 was considered statistically significant.

## 4. Conclusions

In this study, constipation-related parameters, including histopathology and immunohistochemistry, gastrointestinal hormone levels, related gene expression levels, intestinal metabolites, and gut microbiota structure, were analyzed to explore the influences of HLP on constipation induced by loperamide in mice. It was proven that HLP played a role in treating constipation by optimizing intestinal flora composition, thus promoting intestinal peristalsis, relieving intestinal inflammation, and regulating intestinal electrolyte metabolism. Moreover, the HLP-M group was found to significantly alter constipation parameters, maintain colonic epithelial integrity, improve serum gastrointestinal regulation-related peptide levels, reverse changes in colon-related protein expression levels, and optimize intestinal flora structure through three different doses of HLP. In conclusion, the medium dose of HLP was the most effective in the treatment of constipation. These results suggested that HLP offered a new biologic therapy for treating constipation. As one of the vital active components of sea cucumber, more effects of sea cucumber polysaccharides are being further explored. At the same time, this study provides a solid theoretical basis for the clinical treatment of constipation and the high-value development of *Holothuria leucospilota*.

## Figures and Tables

**Figure 1 ijms-24-02553-f001:**
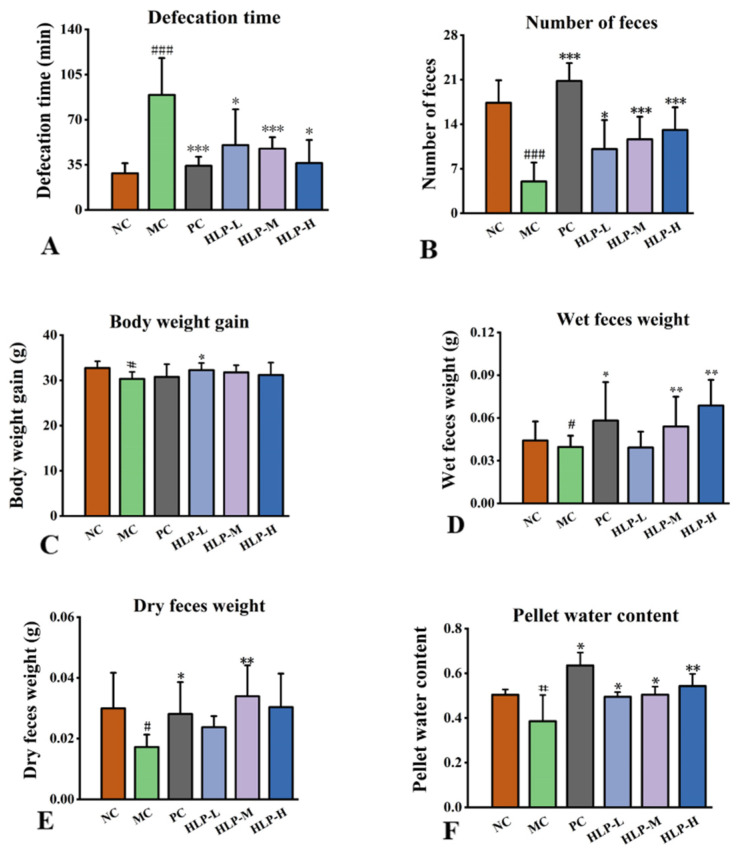
The effects of HLP on the parameters related to defecation in mice. (**A**) Defecation time. (**B**) Number of feces. (**C**) Body weight gain. (**D**) Wet feces weight. (**E**) Dry feces weight. (**F**) Pellet water content. # *p* < 0.05, ### *p* < 0.001 vs. NC group; * *p* < 0.05, ** *p* < 0.01, *** *p* < 0.001 vs. MC group. NC: normal control group, MC: model control group, PC: positive control group, HLP-L: low-dosage HLP group, HLP-M: middle-dosage HLP group, and HLP-H: high-dosage HLP group.

**Figure 2 ijms-24-02553-f002:**
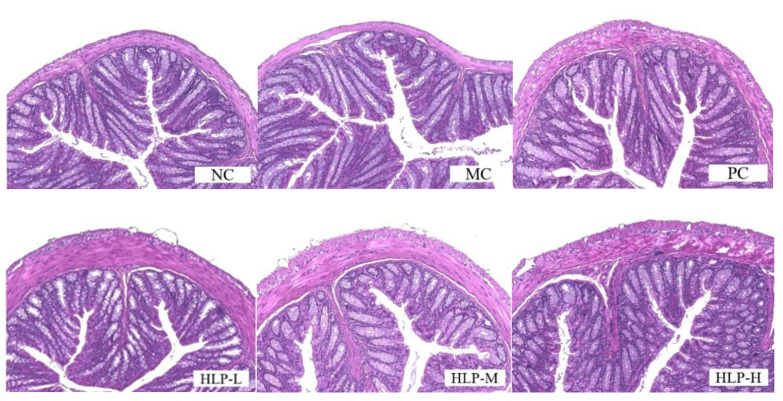
Pathological sections of colon tissue by hematoxylin and eosin (H&E) staining (200× magnification). NC: normal control group, MC: model control group, PC: positive control group, HLP-L: low-dosage HLP group, HLP-M: middle-dosage HLP group, and HLP-H: high-dosage HLP group.

**Figure 3 ijms-24-02553-f003:**
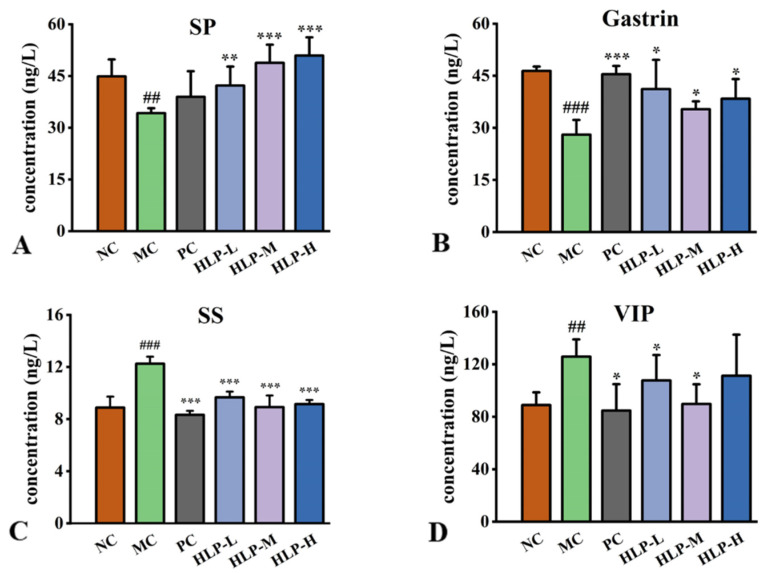
Gastrointestinal regulatory-related peptides content in mice. (**A**) SP: substance P. (**B**) Gastrin. (**C**) SS: somatostatin. (**D**) VIP: vasoactive intestinal peptide. ## *p* < 0.01, ### *p* < 0.001 vs. NC group; * *p* < 0.05, ** *p* < 0.01, *** *p* < 0.001 vs. MC group.NC: normal control group, MC: model control group, PC: positive control group, HLP-L: low-dosage HLP group, HLP-M: middle-dosage HLP group, and HLP-H: high-dosage HLP group.

**Figure 4 ijms-24-02553-f004:**
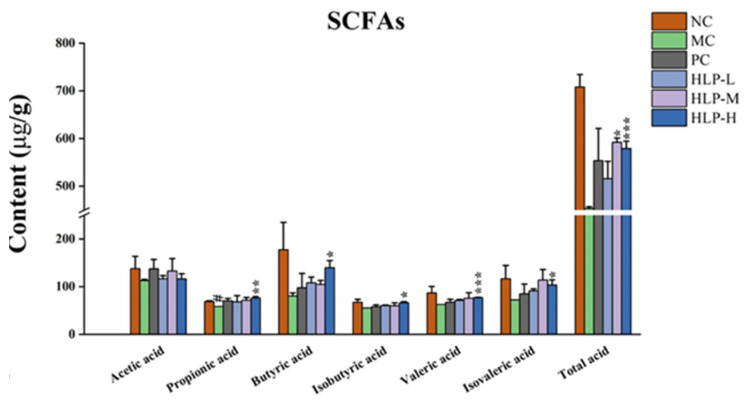
Short-chain fatty acids (SCFAs) content in mice. # *p* < 0.05 vs. NC group; * *p* < 0.05, ** *p* < 0.01, *** *p* < 0.001 vs. MC group. NC: normal control group, MC: model control group, PC: positive control group, HLP-L: low-dosage HLP group, HLP-M: middle-dosage HLP group, and HLP-H: high-dosage HLP group.

**Figure 5 ijms-24-02553-f005:**
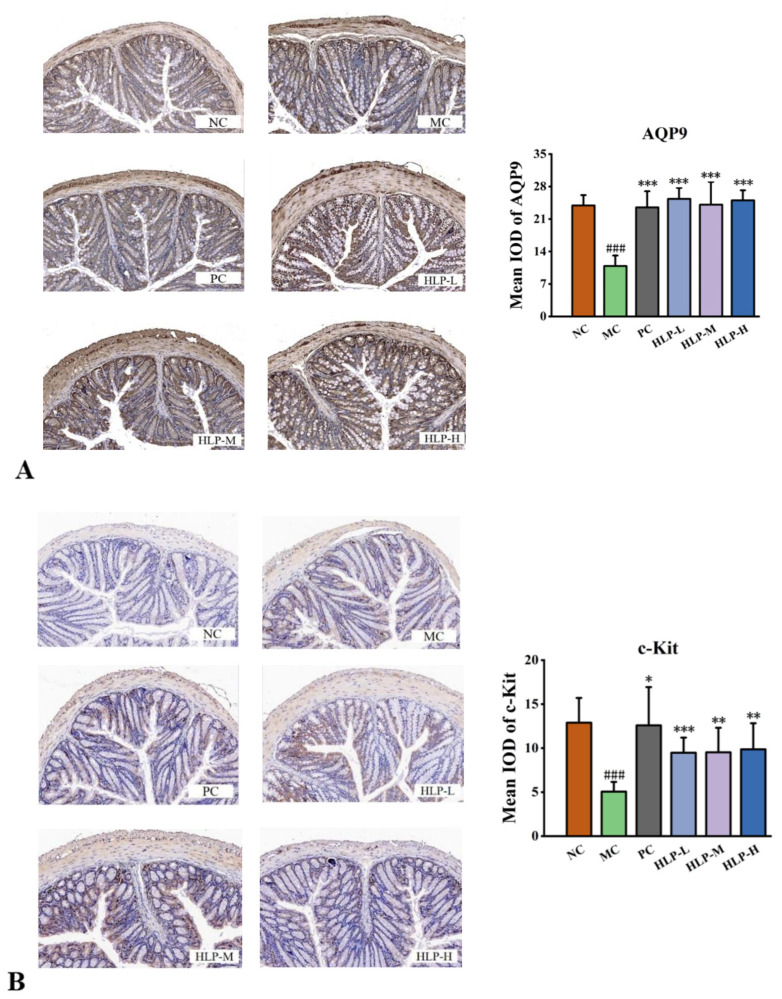
The expression of water metabolism and motility of intestinal tract-related genes and proteins. (**A**) AQP9 immunohistochemistry (200× magnification) and its mean IOD values in colon tissue. (**B**) c-Kit immunohistochemistry (200× magnification) and its mean IOD values in colon tissue. ### *p* < 0.001 vs. NC group; * *p* < 0.05, ** *p* < 0.01, *** *p* < 0.001 vs. MC group. NC: normal control group, MC: model control group, PC: positive control group, HLP-L: low-dosage HLP group, HLP-M: middle-dosage HLP group, and HLP-H: high-dosage HLP group.

**Figure 6 ijms-24-02553-f006:**
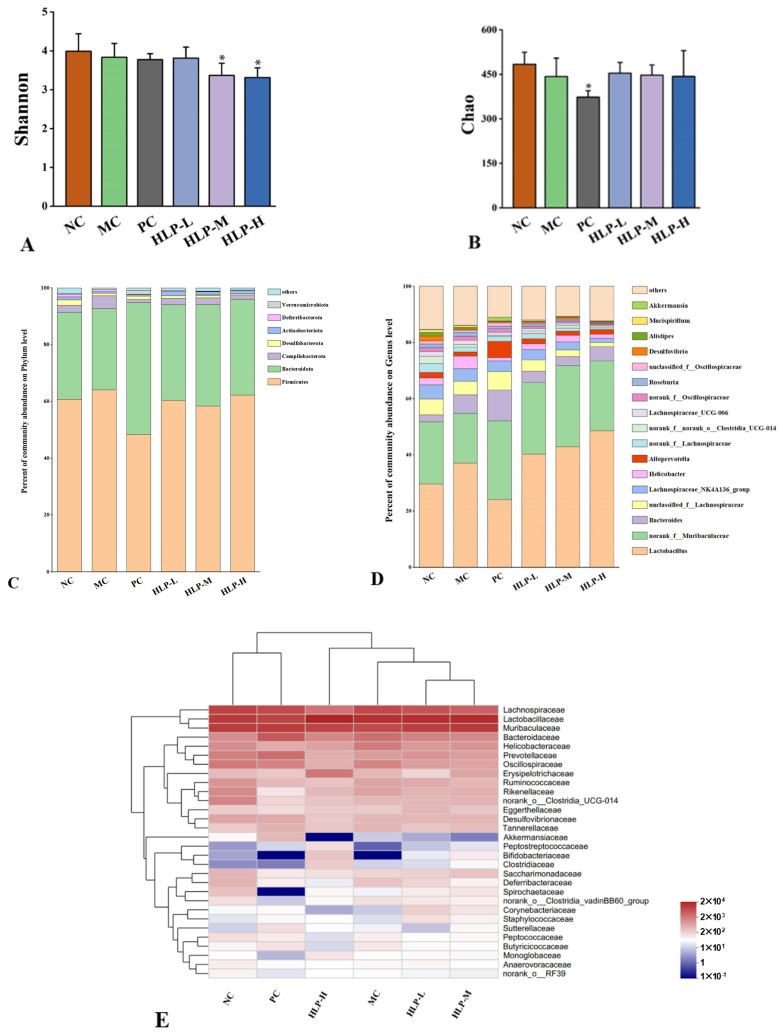
The effects of HLP on microbiota diversity and community structure in mice’s gut. (**A**) Shannon and (**B**) Chao estimations of gut microbiota diversity. * *p* < 0.05 vs. MC group. NC: normal control group, MC: model control group, PC: positive control group, HLP-L: low-dosage HLP group, HLP-M: middle-dosage HLP group, and HLP-H: high-dosage HLP group. Changes of gut microbiota at the (**C**) phylumand (**D**) genus level. (**E**) Heat map comparison between experimental groups at the family level.

**Figure 7 ijms-24-02553-f007:**
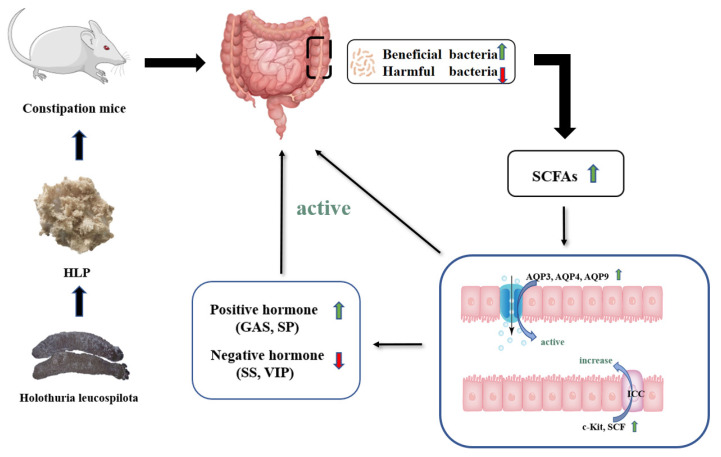
Potential mechanism of HLP relieving constipation in mice.

**Table 1 ijms-24-02553-t001:** Relative expression level of water channel proteins (AQP3, AQP4 and AQP9), c-Kit, and SCF.

Category	Group					
	NC	MC	PC	HLP-L	HLP-M	HLP-H
AQP3	1.20 ± 0.35	0.35 ± 0.13 ^#^	3.19 ± 0.95 **	0.85 ± 0.33	1.15 ± 0.38 *	0.54 ± 0.26
AQP4	1.19 ± 0.36	0.32 ± 0.15 ^#^	1.62 ± 0.80 *	0.75 ± 0.11 *	1.38 ± 1.06	1.12 ± 0.69
AQP9	0.44 ± 0.00	0.16 ± 0.09 ^#^	1.40 ± 0.42 *	0.69 ± 0.11 *	0.57 ± 0.11 *	0.43 ± 0.00 *
SCF	1.03 ± 0.14	0.89 ± 0.05 ^#^	2.32 ± 1.55 *	0.92 ± 0.40 *	4.14 ± 0.58	2.06 ± 0.20
c-Kit	1.02 ± 0.50	0.25 ± 0.05 ^#^	1.36 ± 0.40 *	1.03 ± 0.10	1.12 ± 0.07 ***	1.40 ± 0.38 *

# *p* < 0.05 vs. NC group; * *p* < 0.05, ** *p* < 0.01, *** *p* < 0.001 vs. MC group. NC: normal control group, MC: model control group, PC: positive control group, HLP-L: low-dosage HLP group, HLP-M: middle-dosage HLP group, and HLP-H: high-dosage HLP group.

**Table 2 ijms-24-02553-t002:** Gene Primer of water channel proteins (AQP3, AQP4, and AQP9), c-Kit, SCF and GADPH.

Gene Primer	Primer Sequence
AQP3	Forward 5′-GCTGTCACCCTTGGCATCTTGG-3′ Reverse 5′-AGGAAGCACATTGCGAAGGTCAC-3′
AQP4	Forward 5′-CAGCATCGCTAAGTCCGTCTTCTAC-3′ Reverse 5′-ACCGTGGTGACTCCCAATCCTC-3′
AQP9	Forward 5′-GCTGTCCTGGGAGGTCTCATCTATG-3′ Reverse 5′-GCTGGTTCTGCCTTCACTTCTGG-3′
SCF	Forward 5′-TGCGGGAATCCTGTGACTGATAATG-3′ Reverse 5′-CCGGCGACATAGTTGAGGGTTATC-3′
c-Kit	Forward 5′-GATCTGCTCTGCGTCCTGTTGG-3′ Reverse 5′-AACTCTGATTGTGCTGGATGGATGG-3′
GADPH	Forward 5′-GGTTGTCTCCTGCGACTTCA-3′ Reverse 5′-TGGTCCAGGGTTTCTTACTCC-3′

## Data Availability

All raw data supporting the reported results are available from the authors on request.

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
