# Peer review of "Polysaccharides from Holothuria leucospilota Relieve Loperamide-Induced Constipation Symptoms in Mice"

_ijms, 2023, doi:10.3390/ijms24032553_

Round 1
Reviewer 1 Report
I recommend that the authors should include quantitative results in the abstract section. Also, the keywords should be charged because they are repeated in the title.
The authors omitted some important works that they could include in the introduction section. Below there is a brief list of these papers.
Zhao et al. (2020). A sea cucumber (Holothuria leucospilota) polysaccharide improves the gut microbiome to alleviate the symptoms of type 2 diabetes mellitus in Goto-Kakizaki rats.
Zhao et al. (2022). Holothuria Leucospilota Polysaccharides Improve Immunity and the Gut Microbiota in Cyclophosphamide-Treated Immunosuppressed Mice.
Lu et al. (2022). Holothuria Leucospilota polysaccharides alleviate hyperlipidemia via alteration of lipid metabolism and inflammation-related gene expression.
The figures should include the meaning of the abbreviations in the full description.
Reviewer 2 Report
To improve the quality of the manuscript suggested to go through and update the following comments
Line 55: Durio zibethinus Murr rind & Line 56: Chrysanthemum morifolium polysaccharides are derived from?
Line 275: What is reason for choosing only female mice?
Line 279 – 281 : Author suggests to mentioning the doses in the low-dosage HLP group (HLP-L group), middle-dosage HLP group (HLP-M group) and high-dosage HLP group (HLP-H group), normal control group (NC group), model control group (MC group), positive control group (PC group).
Normal group and Positive control group treated with? mention where mentioned at first.
What is the meaning of model control group (MC group), briefly explain.
Line 288 : 20 mg/mL? check the unit.
mg/g or mg/kg BW? check the unit.
Result : 2.2 : From the result compared to the NC group, HLP groups muscular layer thickness looks like more thickness (like double the volume) what will be the justification?
The author did not discuss anything about 2.1 and 2.2 results.
What is the importance of 16S rRNA gene sequences in gut microbiota, author must explain?
Figure 6 (C & D) description is not readable.
Line 301 : What is the purpose of gavaged with 2 mL of ink.
Line 316 : Author suggested to mention the catalog number with host for respective antibodies.
Line 327 : ELISA kits catalog number?
Line 335 : Eastep®Super Total RNA Extraction Kit catalog number?
Line 338 : First Strand cDNA 338 Synthesis Kit Catalog number?
Line 340 : SuperReal Preix Plusreagent kit (SYBR Green) Catalog number?
How did the author choose the 10 mg/kg BW loperamide? Because loperamide is used for Diarrhea treatment, in this case, 10 mg/kg BW loperamide is high does to make constipation to the mice in this study? Author must clarify.
How did the author choose the HLP concentration? From this study, the author must conclude along with which concentration of HLP is more effective to treat the Constipation Symptoms.
